# Design of Gold Nanoparticle Vertical Flow Assays for Point-of-Care Testing

**DOI:** 10.3390/diagnostics12051107

**Published:** 2022-04-28

**Authors:** Rongwei Lei, David Wang, Hufsa Arain, Chandra Mohan

**Affiliations:** Department of Biomedical Engineering, University of Houston, Houston, TX 77004, USA; rlei3@uh.edu (R.L.); davidwang200@gmail.com (D.W.); harain@uh.edu (H.A.)

**Keywords:** vertical flow assay, gold nanoparticles, lateral flow assay, point-of-care testing, limit of detection (LoD), multiplexing capability

## Abstract

Vertical flow assays (VFAs) or flow-through assays have emerged as an alternate type of paper-based assay due to their faster detection time, larger sample volume capacity, and significantly higher multiplexing capabilities. They have been successfully employed to detect several different targets (polysaccharides, protein, and nucleic acids), although in a limited number of samples (serum, whole blood, plasma) compared to the more commonly known lateral flow assays (LFAs). The operation of a VFA relies mainly on gravity, coupled with capillary action or external force to help the sample flow through layers of stacked pads. With recent developments in this field, multiple layers of pads and signal readers have been optimized for more user-friendly operation, and VFAs have achieved a lower limit of detection for various analytes than the gold-standard methods. Thus, compared to the more widely used LFA, the VFA demonstrates certain advantages and is becoming an increasingly popular platform for obtaining qualitative and quantitative results in low-resource settings. Considering the wide application of gold nanoparticles (GNPs) in VFAs, we will mostly discuss (1) the design of GNP-based VFA along with its associated advantages/disadvantages, (2) fabrication and optimization of GNP-based VFAs for applications, and (3) the future outlook of flow-based assays for point-of-care testing (POCT) diagnostics.

## 1. Introduction

Point-of-care testing (POCT) is a user-friendly, rapid, sensitive diagnostic approach for detecting the presence of targets of interest that is in high demand in resource-limited settings. During the past 50 years, POCT applications in health care, agriculture, food safety, forensic science, animal health, and the military have been developed [1]. Among the different types of POCT, the lateral flow assay is most widely applied across all fields due to its simplicity of operation and rapid reading. In the past ten years, lateral flow assays based on fluorescent nanoparticles (NPs), luminescent NPs, enzymatic reactions, and colorimetric NPs have been developed to meet the requirements for higher sensitivity, accuracy, and multiplexing capability [2]. According to the World Health Organization (WHO) guidelines, POCT should meet the following criteria: affordable, sensitive, specific, user-friendly, rapid, and robust; requiring no complex equipment; and can be delivered efficiently to end-users [3].

Although lateral flow assays are coupled with portable readers such as smartphones and various sensors to obtain quantitative measurements, lateral flow assays (LFAs) are still limited by (i) low multiplexing capacity (<10), (ii) the hook effect (false negative), (iii) low sample volume capacity (<100 µL), and (iv) moderate speed (15–40 min), as summarized in Table 1. For example, rapid commercial tests such as the HIV 1/2 Ag/Ab Combo, the TB LAM Ag test, the Influenza A + B test, the Malaria Pf Test, ImmunoCard STAT! *E. coli* O157 Plus and the multiplex lateral-flow assay RAIDTM 5 to detect biological threat agents are all designed for the detection of a limited number of targets [1,2,4]. A typical LFA strip consists of four overlapping elements mounted on an adhesive backing. The first element is the sample pad, typically made of cellulose or glass fiber to introduce the sample of interest to the second element, the conjugate pad at a constant rate. The conjugate pad is typically made of cellulose, glass fiber, or polyesters depending on the type of labeled conjugates and the assay’s sensitivity [2,5]. The labeled biomolecules are stored in the conjugate pad and should bind to the analyte in the sample of interest when the sample of interest reaches the conjugate pad. The analyte-conjugate complex laterally flows through the third element, typically the nitrocellulose membrane, where specific biological compounds (typically antibodies, protein, or nucleic acids) are immobilized at pre-defined lines. The analyte, analyte-conjugate complex, and conjugates should react specifically to the compounds dispensed on the membrane. Lastly, the fourth element, the absorbent pad, should absorb any remaining sample of interest and conjugate complex [2,5]. Depending on the molecular weight or structure of the analyte of interest, LFAs can be majorly classified into two categories: one is LFA, where antibodies are used as recognition elements to detect proteins, the other is nucleic acid lateral flow assay (NALFA), where nucleic acids are used as recognition elements to detect amplicons or results of amplification reactions like polymerase chain reaction (PCR) or recombinase polymerase amplification (RPA) [1,2]. 

Due to the intrinsic nature of antibody-antigen immunoreaction on each test line, the physical distance of multiplexing test lines on a single lateral flow strip has hindered operators from incorporating multiple analytes on one strip [5,6,7]. Additionally, crosstalk or poor sensitivity is usually seen in the antibody-based detection of more than four analytes due to antigenic similarity and compromised buffer conditions. Therefore, it has been particularly challenging to develop multiplexed LFAs on a manufacturing scale into commercial products [2]. Although researchers have explored multiple approaches for the simultaneous detection of up to 10 targets by fabricating a star shaped two-dimensional LFA [8] and have assessed multiplex detection on a single test line [9] by having multiple capture antibodies on a single line, these approaches are limited in how many targets they can interrogate. The hook effect was also observed in LFAs due to the mixing of reporter and sample, causing false negatives [2]. Nevertheless, paper-based lateral flow assays have proven their utility in detecting analytes in a wide range of clinical samples, as shown in Table 1. However, the small volume capacity of such assays has limited the sensitivity, especially for low-level analytes. Clinical samples with high viscosity usually necessitate dilution to facilitate the flow, further diluting the already low levels of each analyte, thus amplifying the difficulty of detecting a signal in the LFA [2].

Another issue with LFAs is their moderate assay response time between 15–40 min. Take for instance LFAs for the COVID-19 pandemic that has struck worldwide since 2020. Compared to the gold standard of real-time PCR, an increasing number of LFA-based diagnostics have earned emergency use authorization (EUA) from the Food and Drug Administration (FDA), under the names COVID Rapid Antigen Test or COVID Serology Test [10,11,12]. These tests, which focus on detecting SARS-CoV-2 nucleoprotein or IgG/IgM against SARS-CoV-2, usually take 15 min to 30 min for a readout. Indeed, these are “rapid” compared to the hours required for PCR, but not rapid enough for diagnosis by health care providers facing incessant waves of infected patients. Additionally, with the discovery of increasing numbers of mutated strains of SARS-CoV-2, it is difficult to identify the early stages of SARS-CoV-2 outbreaks by targeting antigens or antibodies to one or two specific strains, especially in resource-limited settings [13]. In 2021, growing interest in rapidly differentiating the symptoms of fever, cold, and headache caused by COVID-19 from those caused by other inflammatory diseases has created a high demand for an assay capable of rapid multiplex detection [14,15].

An alternative to LFAs is rapid vertical flow assays (VFAs), which offer several advantages, including faster response (1–40 min), no timing requirement (signal maintained for hours after the completion of the assay), high multiplexing capacity (>1000), the absence of a false-negative-inducing hook effect, and high sample volume capacity (>500 µL) [16]. As a result, many next-generation POCT diagnostics are beginning to explore VFAs [17,18]. In recent decades, VFAs with porous membranes have been utilized for the multiplex detection of nucleic acids, proteins, antibodies, polysaccharides, and virus antigens with greatly increased multiplexing capacity in microarray format, as well as increased sensitivity with multi-stacked protein microarrays coupled with filtration. By combining gravity, external forces, and capillary forces, VFA is usually faster than LFA. Furthermore, due to the separation step between sample loading and reporter loading, VFAs avoid the hook effect, which favors the detection of highly concentrated analytes. Finally, VFAs feature high sample volume capacity, potentially increasing the limit of detection of analytes and allowing room for a higher dilution factor of clinical samples [16].

Despite the relative advantages of VFAs over LFAs, VFAs are still new, and many reported studies do not fully meet the WHO-issued criteria [3]. Many comprehensive reviews on LFAs have been reported [2,19,20,21]; however, no comprehensive work focusing primarily on the application of VFAs exists. Hence, this review discusses VFA applications in medical diagnostics, including the detection of cancer, cardiovascular disease, HIV, and bacterial and viral infection-related biomarkers. The studies included in this review are based on colorimetric detection using gold nanoparticles (GNPs), surface-enhanced Raman scattering (SERS), enzymatic reactions to provide qualitative or semiqualitative readouts, and even quantitative measurement enabled by a smartphone. Moreover, this review includes details on multi-stack materials, buffer systems, diagnostic issues, and potential solutions to facilitate the fabrication and optimization of vertical flow assays.

## 2. Materials and Methods

PubMed search was conducted on 1 August 2021 to gather relevant papers, using the following search strings: “Vertical flow assay”, “Flow through”, “point of care”, separately. Vertical flow assay studies that met at least one of the World Health Organization (WHO) point of care testing (POCT) criteria were selected for review. Following these steps, a total of 56 studies that met these criteria were included in this review as shown in Figure 1.

## 3. Results

### 3.1. VFAs in POCT Diagnostics

Vertical flow assays (VFAs) operate mainly through gravity, where the sample flow is vertical or perpendicular to the paper, and partially through external force and capillary action [16]. On the other hand, lateral flow assays (LFAs) consist of a sample flow parallel to the paper’s surface, permitting wicking only by capillary action [2,6]. This has limited the sensitivity and multiplexing capability of LFAs since the uniform flow rate of LFAs requires pore sizes of several micrometers (5 µm to 15 µm), and rapid response requires a limited length (5 cm to 20 cm) of the membrane [1,2,6]. However, LFAs and VFAs share similar fundamental principles, as they both work by immobilizing a capture substrate onto a reagent pad and applying a sample (in the presence or absence of the target analytes) [16]. First, a typical VFA membrane (NCM) is cut and separated into two zones: the control zone “C” as a fail-safe and the test zone “T” as a signal indicator. Then, the NCM is assembled on top of multiple stacked absorption pads and immobilized with a capture antibody. Next, the NCM is prewetted and loaded with the sample. After absorption, reporters such as gold nanoparticles (GNPs) are added to the NCM, along with several washes. In VFAs, the interactions between the specific antigen, capture antibody, and GNPs result in an immediate and permanent colored dot that can be detected by naked eye or with a smartphone reader, as Figure 2 shows. The main biomarkers currently detected in VFA applications are antibodies, protein antigens, and nucleic acids, as listed in Table 2, Table 3, Table 4 and Table 5 [22,23,24,25,26,27,28,29,30,31,32,33,34,35,36,37,38,39,40,41,42,43,44,45,46,47].

The diagram shows the commonly adopted protocol for vertical flow assays. First, the membrane is pre-wet with the wash buffer (optional). Next, the membrane is blocked; blocking time varies based on the researcher’s discretion. The blocking buffer is washed away (optional) and test samples are added to the membrane. The membrane is washed (optional) and gold nanoparticles are added; the reporter could be antibody-functionalized for direct color observation or a color amplification step can be introduced using a biotin-streptavidin link. Once the sandwich assay is completed, one final washing step follows (optional), and the results are visualized.

### 3.2. Antibody Detection

Thus far, MedMira Inc. is the only company that has launched a rapid vertical flow assay (VFA) detection kit, offering a comparable but faster alternative to the corresponding lateral flow assay [28]. Based on gold nanoparticle (GNP) detection, this manufacturer has developed multiple VFAs for detecting viral, bacterial, or fungal infections (e.g., HIV-1/HIV-2, HBc/HIV/HCV, etc.) as well as blank VFAs that can be customized for multiplex screening of other targets (https://medmira.com/wp-content/uploads/2018/01/Miriad-RVF-Toolkit-Product-Sheet_EN.pdf, accessed on 15 January 2021). 

Compared to standard-GNP-based VFAs, where reddish colors are visually analyzed and potentially suffer from limited precision and sensitivity, other VFA studies have investigated surface-enhanced Raman scattering (SERS) due to its advantages such as photostability, non-destructiveness, and ultra-sensitivity at the single-molecule level [24,29,36,38,40,48]. One group reported a SERS-based VFA for measuring antibodies against hepatitis C with a visual detection limit of 63.1 µg/mL, digital detection limit of 3.32 µg/mL, 1 min assay response time, and low spot-to-spot variation [29]. Another SERS-based rapid VFA study reported a detection limit of 3 ng/mL of mouse IgG within a 2 min assay time [24]. Reutersward and Chinnasamy et al., two groups, reported a GNP-based VFA to detect IgE for hyper IgE syndrome and allergens, respectively [34,35]. Both reports showed that a VFA with high multiplex capabilities (>1000 immunoreaction spots) could be used to detect serum IgE, exhibiting higher sensitivity than the standard ImmunoCAP assay. 

Since 2020, SARS-CoV-2 has spread dramatically with a colossal death toll worldwide, calling for a rapid diagnostic testing to complement real time-polymerase chain reaction (RT–PCR). One recent work described using an HRP/TMB-based VFA for the rapid detection of SARS-CoV-2 antibodies [27]. In this work, they proposed using multiple membrane layers for a more constant flow rate. SARS-CoV-2 nucleocapsid protein (NP) was fused with cellulose binding domain (CBD) as the capture reagent for higher binding efficiency. Instead of immobilizing NP-CBD, they used the target antibody mixed with NP-CBD and NP-biotin-streptavidin horseradish peroxidase conjugate (NP-HRP) as the reporter reagent. Results demonstrated a cyan color signal after adding TMB, corroborated using a chemiluminescence immunoassay [27]. 

Smartphones with high-resolution rear cameras are emerging as a valuable tool for POCT diagnostics, serving as a detection apparatus for colorimetry, fluorescence, and luminescence [4]. With appropriate software, they can also be used for imaging, data processing, storage, and communication. This could be extremely useful for patients who need long-term monitoring of their disease status and for providing physicians with information to guide early treatment intervention. Smartphone-based LFAs are becoming common, but remain new for VFAs. To date, only one study has reported multiplex detection coupled with a mobile phone to detect antibodies against MbpA, OspC, and P41 for Lyme disease [31].

### 3.3. Protein Biomarker Detection

Raman dye-encoded core-shell (Au@Ag) nanostructure-based surface-enhanced Raman scattering (SERS) nanotags have been synthesized and applied to multiplex bioassays to obtain high-throughput bioinformatics data. Chen et al. reported a SERS-based VFA where one test zone supported the detection of three analytes: Alpha-fetoprotein (AFP), Carcinoembryonic Antigen (CEA), and prostate-specific antigen (PSA), for prostate cancer [36]. This rapid assay (10 min) showed a limit of detection (LoD) lower than 1 pg/mL for all three targets, indicating that the combination of SERS and VFA has tremendous potential for biomarker analysis and disease diagnosis. Another SERS nanotag-based VFA study reported ultrasensitive detection of inflammatory biomarkers, achieving multiplex detection of C-reactive protein (CRP), interlukin-6 (IL-6), Serum amyloid (SAA), and procalcitonin at the fg/mL level [38]. Notably, the SERS reader is portable but several times larger than smartphones or other readers used in LFAs, limiting their utility for home use. 

Several lateral flow assays (LFAs) have reported rapid and sensitive results for assaying individual targets in cardiovascular disease [48,49]. Compared to LFAs, three VFA studies showed a faster response time [22,23,25], with one study showing a gradual concentration-dependent increase in signal intensity without any hook effect [23]. Prajapati et al. developed a 1–2 min rapid assay feasible for use with serum/whole blood for CRP detection. However, this semiquantitative assay may have high reader-to-reader variability [25]. Park & Park developed a 3D paper-based microfluidic device to achieve delayed flow to detect CRP [22]. Oh et al. developed a multi-stacked assay in which the sample pad, flow control film, conjugate pad, asymmetric membrane, and nitrocellulose membrane (NCM) were successively assembled [23]. Both of these approaches achieved a one-step assay, in contrast to most VFA studies which require multiple steps to load reagents.

### 3.4. Nucleic Acid and Small Molecule Detection

Recombinase polymerase amplification (RPA) has recently gained popularity as a convenient isothermal nucleic acid test for diagnostics. RPA is relatively fast with an amplification time of 20 min, whereas other amplification methods require hours. Additionally, RPA has one of the lowest operating temperatures (37–42 °C). Many RPA-based LFA studies have reported success in detecting viruses such as SARS-CoV-2 [50,51,52]. Among RPA-based VFA studies, two groups reported their work in detecting adenoviral infection (viral) [26] and meningitis infection (bacterial) [45], both demonstrating high multiplexing capability with spatially separated immunoreaction spots for multiple strain detection. The sample was pushed through a single membrane in steps, allowing the target to react with the membrane. However, DNA extraction is still necessary before amplification and subsequent single-stranded DNA preparation complicates assay operation. These hurdles render RPA less compatible with POCT, warranting an additional on-site ancillary protocol. VFA-based detection of even smaller molecules than DNA, such as iron [46] and oxytetracycline [47], have also been reported. Clearly, the application of VFAs for detecting nucleic acids and metabolites is likely to grow and evolve in the coming years.

### 3.5. Options for Fabrication and Optimization of VFAs

#### 3.5.1. Increasing the Flow Rate and Decreasing Membrane Pore Size Is a Good Combination for Improving VFA Assay Sensitivity

Chen et. al [37] researched a vertical flow assay (VFA) for detection of Burkholderia pseudomallei surface capsular polysaccharide and found that sensitivity was improved fivefold by increasing the flow rate (to 1.06 mm s^−1^) and decreasing the pore size (to 0.1 µm). Some commonly used nitrocellulose membranes with different pore sizes are as follows: Cytiva Amersham Protran Western blotting membrane (0.1 µm, 0.2 µm, and 0.45 µm), MDI membrane technology (0.3 µm, 0.45 µm, 0.8 µm, 5 µm, 8 µm), Sartorius (0.22 µm, 0.45 µm), Bio-Rad (0.45 µm), and BioTraceTM NT Nitrocellulose Transfer Membrane (0.2 µm). The size of the pore is critical for initial screening since it determines the limit of detection (LoD) of the assay and the feasibility for use with different sample types. A more viscous sample requires a larger membrane pore size to maintain a constant flow rate and more homogenous color precipitation. Kim et al. [27] have reported that more constant flow rate was noticed with higher number of membrane layers, but slower flow rate. In addition to the external force from the syringe, membrane porosity, and inclusion of multiple membrane layers, absorption pad capacity is also essential for controlling the flow rate. Some absorption pads with different thicknesses are as follows: Whatman Grade 707 Blotting Pad, Bio-Rad Thick Blot Absorbent Filter Paper, MDI membrane technology (AP080, AP110, AP120), Pall Absorbent Pad Kits and Whatman Dipstick Pad and Papers (CF5, CF7). Generally, higher thickness of absorbent pads has higher assay volume capacity. However, clamping several layers of less thick absorbent pads can also increase the assay volume capacity. 

#### 3.5.2. Buffer Optimization Is an Important Step in Assay Optimization and Troubleshooting

Blocking components such as bovine serum albumin (BSA) and non-fact dry milk (NFDM) are commonly used for reducing non-specific binding. Increasing the blocking component concentration and blocking time can efficiently reduce the background. Standard washing buffers include phosphate buffered saline (PBS), Tris-buffered saline (TBS), and phosphate buffer (PB). With a detergent such as Tween 20, the washing step can be more efficient. The diluent used to dilute the nanoparticles (NPs) is the most crucial buffer because the ultimate signal is highly dependent on NP dispersion and precipitation on the membrane. Sucrose, polyvinylpyrrolidone (PVP), polyethylene glycol (PEG), and Tween are good homogenizers to facilitate the flow, although some empirical testing is necessary to find the optimal size of polymers and detergent types.

#### 3.5.3. Altering the Reporter Is Another Strategy to Enhance the Limit of Detection (LoD)

(1) Optimizing the size (and color) of the gold nanoparticles (GNPs): NPs are sized from 5 nm to 200 nm corresponding to different color options. Based on the selected membrane, pore size, and source of GNPs, GNPs of different sizes/colors can exhibit widely different performance with respect to signal and flow rate. Hence, various combinations have to be tested in order to optimize the assay. (2) Use of other nanoparticles: Mehta et al. used microbially synthesized silver nanoparticles (AgNPs) that changed from light brown/yellow to dark brown to yield a LoD of 1 × 10^−5^ mM cysteine, with a two-minute turnaround time [39]. Many fluorescent NPs (quantum dots (QDs), up-converting nanoparticles (UCNPs)) are 1–100 nm in size, rendering them good candidates for next-generation vertical flow assay diagnostics. (3) Enzymatic reaction: HRP- or AP-based color signals are sensitive and free from concerns relating to membrane clogging. By combining GNPs with HRP, an enhanced signal can be expected compared to AuNPs or HRP-TMB alone.

## 4. Discussion

Most of the reported vertical flow assays (VFAs) only meet some of the world health organization (WHO)-issued criteria: affordable, sensitive, specific, user-friendly, rapid, robust, requiring no complex equipment, and can be delivered efficiently to end-users [3]. To improve the utility of VFA, conjugate immunopads [32] and integrated conjugate pads [23] should be coupled with the assay for one-step operation. Sample and reagent dilution/preparation are a long and complicated process for most users. For example, clinical samples such as blood or saliva need pretreatment to prevent clogging the membrane. Thus, a simplified sample and reagent loading protocol should be incorporated. In the long run, quantitative measurements by smartphones or truly portable readers will play an increasing role in at-home diagnosis. Despite the enhanced sensitivity and multiple color availability, the cost of commercial Raman imaging instrumentation is high, and its size is several times larger than that of a smartphone; thus, this technology needs to be simplified. Enhanced VFA performance in conjunction with fluorescent nanoparticles (NPs), luminescent NPs, and colorimetric NPs will lead to a new category of analytical devices in the diagnostics arena, moving disease diagnostics to the home.

## Figures and Tables

**Figure 1 diagnostics-12-01107-f001:**
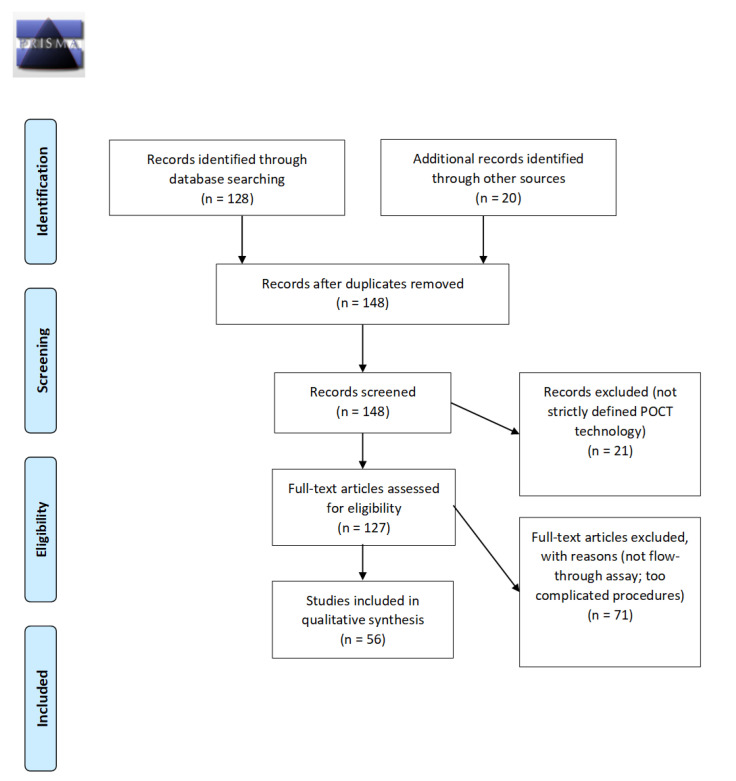
PRISMA flow chart of search methods used in this study. PubMed search was conducted on 1 August 2021 to gather relevant papers, using the following search strings: “Vertical flow assay”, “Flow through”, “point of care”. Vertical flow assay studies that met at least one of the World Health Organization (WHO) point of care testing (POCT) criteria were selected for review. Following these steps, a total of 56 studies that met these criteria were included in this review.

**Figure 2 diagnostics-12-01107-f002:**

Commonly adopted protocol for vertical flow assays.

**Table 1 diagnostics-12-01107-t001:** Comparison between LFAs and VFAs for POCT diagnosis.

Features	LFAs	VFAs
Sample flow	Capillary force [6]	External force; Gravity force; Capillary force [16]
Flow method	Passive [6]	Passive; Active [16]
Sensing response	Moderate [6]	Fast [16]
Washing steps	Not required [6]	Mostly yes [16]
Timed results	Required [6]	Not required [16]
Hook effect	Yes [2,6]	Mostly No [22,23]
Sample and conjugate separation	Mostly No [2,6]	Yes [24,25,26] No [22,23,27]
Sample volume	<100 µL [2,6]	10–500 µL [22,23,24,25,26,27,28,29,30,31,32,33,34,35,36,37,38,39,40,41,42,43,44,45,46,47]
Sample type	Urine; Serum; Blood; Plasma; Sweat; Mucus; Saliva; Stool; Food; Cerebrospinal fluid [1,2,5]	Serum; Blood; Plasma [22,23,24,25,26,27,28,29,30,31,32,33,34,35,36,37,38,39,40,41,42,43,44,45,46,47]
Reagents volume	<100 µL [2]	<10 mL [22,23,24,25,26,27,28,29,30,31,32,33,34,35,36,37,38,39,40,41,42,43,44,45,46,47]
Multiplexing capacity	<10 [8,9]	>30 [30,35,43,45]
Detection method	Fluorescent NPs (QD; UCNP); Luminescent; NPs (Phosphors); Enzymatic reaction (HRP); Colorimetric NP (AuNP; CNP/CNT; Latex beads; MNP) [2]	Mostly colorimetric NPs (AgNPs; AuNP; SERS-AuNP) [16]; Enzymatic reaction (HRP; AP) [27]
Measurements	Qualitative or Quantitative coupled with portable reader [2,4]	Mostly qualitative or quantitative coupled with benchtop scanner [16]

Abbreviations: QD: Quantum Dot, UCNPs: Upconverting Nanoparticles, HRP: Horseradish Peroxidase, CNT/CNT: Carbon Nanoparticle/Carbon Nanotube, MNPs: Magnetic Nanoparticles, SERS: Surface-enhanced Raman Spectroscopy, AP: Alkaline Phosphatase.

**Table 2 diagnostics-12-01107-t002:** Vertical flow assays for antibody detection.

Analyte (Ref)	Site of Use (Intended)	Indications	Detection Method	Time	Sample Type	LoD	Advantages	Disadvantages
Antibodies to HIV-1 and HIV-2 [28]	Clinic	HIV	GNP-colorimetric	5 min	Serum, plasma, and venipuncture or fingerstick whole blood specimens	N/A	Sensitivity of 99.8% and a specificity of 99.7%.	Three-steps and expensive
Anti-HCV IgG [29]	Lab Clinic	Hepatitis C	GNP + SERS	1 min	Commercial solution of monoclonal antibodies	Visual/SERS limit 63.1 and 3.32 µg/mL	Reproducible and intense results, little spot to spot variation	Raman spectrometer needed
Anti-LPS O9 IgM [30]	Clinic	Typhoid Fever	GNP-colorimetric	30 min	Plasma	N/A	Multi-well multiplex assay; positive and negative controls can be performed in the same well as samples	Preparation of samples takes longer (~2 h), centrifuge and flatbed scanner required
Anti-MbpA IgG [31]	Clinic Home	Lyme disease	GNP-colorimetric	20 min	Serum	162.2 ng/mL	Mobile-phone based quantitative assay, inexpensive, no advanced medical equipment needed, no trained technicians needed, no need for sample dilution	Requires setup of a central server to interpret results, further studies needed for stability of test + buffers long term
Anti-OspC IgG [31]						209.6 ng/mL		
Anti-P41 IgG [31]						1.05 μg/mL		
Brucella antibodies [32]	Lab	Brucellosis	GNP-colorimetric	5 min	Serum	17:40	98% accuracy, potentially suitable for testing whole blood	Requires heating serum sample to 56 Celsius, samples must be incubated overnight
COVID-19 total Antibody [33]	Clinic	COVID-19	GNP-colorimetric	3 min	Serum, plasma or whole blood samples	N/A		Three-steps and expensive
SARS-CoV2 Antibody [27]	Clinic	COVID-19	NP-Biotin-Streptavidin-HRP colorimetric	10 min	Serum	0.5 nM	Cheap, rapid, easy to operate	Qualitative, in need of scanner for intensity analysis
IgE [34]	Clinic	Hyper IgE syndrome	GNP-colorimetric	8 min	Serum	1.9 μg/mL	Inexpensive; Simultaneously screen 113 samples where 1208 spots are available.	Overnight drying after serum sample added; flatbed scanner needed for detection
IgE-reactive allergens [35]	Clinic	Allergies	Neutravidin-GNP colorimetric	15 min	Serum	1 ng/mL IgG	Large multiplex capabilities (1480 spots available), low CV	Tabletop scanner and software required, skilled labor
mouse IgG [24]	Clinic	NA	SERS GNC-colorimetric	2 min	Serum	3 ng/mL	A plasmonic filter paper with pre-adsorbed goat anti-mouse IgG antibody to improve sensitivity; less than 2 min assays time	using FEI-Quanta 450 SEM to image; Enwave Optronics, Inc. ProRaman-L-785B instrument to analyze

Abbreviation: HCV: Hepatitis C Virus, HIV: Human Immunodeficiency Virus, LPS: Lipopolysaccharide, MbpA: Mannosebinding Protein A, OspC: Outer Surface Protein C, SARS: Severe Acute Respiratory Syndrome.

**Table 3 diagnostics-12-01107-t003:** Vertical flow assays for protein detection.

Analyte (Ref)	Site of Use (Intended)	Indications	Detection Method	Time	Sample Type	LoD	Advantages	Disadvantages
AFP [36]	Clinic	Prostate cancer	SERS	10 min	Serum	0.26 pg/mL	Sensitive, rapid, one test zone to show three detections	Surface-enhanced Raman scattering (SERS)-based vertical flow assay in need of Raman microscope
CEA [36]						0.43 pg/mL		
PSA [36]						0.37 pg/mL		
Capsular polysaccharide [37]	Lab	*B. pseudomallei* (meliodosis)	GNP-colorimetric	<30 min		0.02 ng/mL	Analyte concentration is highly consistent with signal intensity	Sample type unspecified. Assumed to have used purified *B. pseudomallei*, results may not be similar for whole blood/serum testing
CRP [25]	Clinic Home	Asymptomatic cardiovascular disease	GNP-colorimetric	1–2 min	Serum/Whole blood	10 ng/mL	Inexpensive, highly accurate, no expensive equipment needed to interpret results	Semi-quantitative; signal intensity of low-risk CRP concentrations (<1 mg/L) looks similar to high-risk CRP concentrations
CRP [22]	Clinic	Cardiovascular disease		15 min	Serum	0.005 µg/mL	Upper limit of 5 ug/mL without hook effect, one-step assay (no need to sequentially add reagents)	Low sample volume capability; need of benchtop scanner for quantitative measurement
CRP [23]	Clinic	Cardiovascular disease	GNP-colorimetric	2 min	Serum	10 ng/mL	One-step assay (no need to sequentially add reagents), upper limit of 10 ug/mL without hook effect	No mentioning on the stability; need of benchtop scanner for quantitative measurement
CRP [38]	Clinic	Inflammatory biomarker detection	Raman dyes encoded core–shell SERS nanotags	NA	Serum	53.4 fg/mL	Ultra-sensitive; a linear range spanning five orders of magnitude; The proposed method shows acceptable accuracy and repeatability	In need of Raman spectrum measurement system to analyze; In need of longer assay time compared to other VFA.
IL-6 [38]						4.72 fg/mL		
SAA [38]						48.3 fg/mL		
Procalcitonin [38]						7.53 fg/mL		
Cysteine [39]	Clinic Home	Cystinuria	AgNP-colorimetric	2 min	Standard cysteine solution	10 nM visible limit; 0.1 nM quantification limit	Inexpensive, no skilled labor or sophisticated equipment, specific assay	Semi-quantitative, some normal ranges of cysteine concentration look visually positive
Flucytosine [40]	Clinic	Therapeutic drug monitoring	SERS	15 min	Serum	10 μg/mL	No serum dilution needed	Raman spectrometer needed
hCG [41]	Clinic	pregnancy	GNP-colorimetric	10 min	urine	0.5 mIU/mL	Cheap, rapid, small volume requirement (20 uL)	Need to peel the device to see the readout; need washing compared to commercial lateral flow test
HINI [42]	Clinic	Influenza	Electrochemical and colorimetric	6 min	saliva	4.7 PFU/mL in saliva by EIS, 2.27 PFU/mL in saliva by colorimetric	High sensitivity, simplicity of operation; duo-methods detection with higher accuracy	Electrochemical signal needs electrochemical impedance spectroscopy (EIS) for measurements; reduced signal after 30 days storage
HIV p24 and hepatitis B virus antigens [43]	Clinic	HIV and Hepatitis B	Streptavidin-AP + NBT/BCIP	5 min	Pure bovine serum	0.95 ng/mL for HIV p24 and 1.12 ng/mL for HBV-SAg	Multi-well multiplex, sensitive	Higher background in serum sample Need benchtop to analyze the signal
MMP-8 [44]	Lab Clinic	Periodontal disease	Eosin-based signal polymerization, colorimetric	N/A		~1 nM 1.1 nM	75% recovery of saliva sample after processing is finished, only about 15 μL original sample needed	Signals near the lower end of clinically relevant concentrations not easily discernible, qualitative, saliva must be centrifuged and frozen
MMP-9 [44]								

Abbreviations: AFP: Alpha-fetoprotein, BCIP: 5-bromo-4-chloro-3-indolyl-phosphate, CEA: Carcinoembryonic Antigen, CRP: C-reactive Protein, IL-6: Interleukin-6, MMP: Matrix Metalloproteinase 9; NBT: Nitro Blue Tetrazolium, SAA: Serum amyloid A, PSA: Prostate-specific Antigen.

**Table 4 diagnostics-12-01107-t004:** Vertical flow assays for nucleic acid detection.

Analyte (Ref)	Site of use (Intended)	Indications	Detection Method	Time	Sample Type	LoD	Advantages	Disadvantages
Adenoviral DNA [26]	Lab Clinic	Adenoviral infection	GNP-colorimetric	6 min	Constructed amplicons + nasopharyngeal aspirates from patients	50 nM	Detect multiple strains of adenovirus w/low inter- and intra-assay variation	1 h RPA and ssDNA generation; possible cross-reactivity
DNA [45]	Clinic	*Neisseria meningitidis* (meningitis)	Streptavidin-GNP colorimetric	20 min	Constructed amplicons	38–2.1 × 10^6^ copies/assay	Multiplex detection based on different capture probes	DNA extraction, RPA, ssDNA generation required

Abbreviations: RPA: Recombinase Polymerase Amplification, ssDNA: Single-stranded DNA.

**Table 5 diagnostics-12-01107-t005:** Vertical flow assay for small molecules detection.

Analyte (Ref)	Site of Use (Intended)	Indications	Detection Method	Time	Sample Type	LoD	Advantages	Disadvantages
Iron [46]	On-site clinic	NA	NA	NA	Whole blood	NA	the system consists of a smartphone and an in-house developed app	NA
Oxytetracycline [47]	On-site clinic	Drug abuse	Silver-enhanced GNP-colorimetric	4 min	Fish tissue	2 ng/mL	Simple, sensitive and rapid assay. Room for several test samples.	Multiple steps and lack of reference for semiquantitative measurements.

## Data Availability

Not applicable.

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
