# Peer review of "Design of Gold Nanoparticle Vertical Flow Assays for Point-of-Care Testing"

_diagnostics, 2022, doi:10.3390/diagnostics12051107_

Round 1

Reviewer 1 Report

General comment:

This work deals with the review of VFAs design for point-of-care testing. The focuse is on gold nanoparticles.

Specific comments throughout the paper:

Affiliation

There is a single affiliation, with four email addresses. Please revise. 

Abstract.

The abstract does not mention gold nanoparticles. There is a strike between the title and the abstract. Please revise the abstract.

1. Introduction

Line 28: POCT is defined in the abstract but not in the Introduction section text. Please fix.

Line 34: NP is not defined as an acronym. Please define it.

Lines 40-42: I suggest to number the list of limitations as i), ii) etc, or as a bulleted list. Indeed, there can be confusion, such as in the case of the "multiplexing capacity (<10), 2)". This is an editorial observation. 

Lines 43-47: Why listing explicitely the name of all these commercial tests? 

Lines 60-64: Missing references for these sentences. 

Lines 65-79: This part about COVID is appreciated and demonstrate that the manuscript is timely, while potentially being of interest to a large community of readers, such as that of Diagnostics. 

Table 1 is really relevant for the readers' understanding. Thank you for providing it.

2. Materials and Methods

The review methodology is explained. A point has to be clarified: in line 112, were the keywords for the search used in combination? This information should be reported. 

3. Results 
3.1 

Lines 124-135: Missing reference(s). 

Tables 2 - 5: I suggest to change the references from "(44)" to "[44]" in the first table column. There should be a coherent notation. 

3.2 

Lines 177-184: Missing reference for the MedMira product. Provide a website or a manual. 

Lines 185-188: Missing reference.

This section is strongly related to one company. The authors did not declare any conflict of interests, but the lack of comparison with other stakeholders and companies is not appropriate.

3.3

Lines 218-219: Too many undefined acronyms. PLease fix. 

3.5

The sub-subsections has to be better highlighted (title, font, etc.) for easing the readers' comprehension. 

Line 256: Extra "size". Please correct this typo. 

Lines 256-274: The text is not justified. Please revise the editing according to the template. 

A note: Maybe some insight about the fundamentals of the physics and biochemistry of LFAs could be of help for the non technical readers who could read your review work and use it as start for their research (e.g., M.Sc. and PhD. students)

Author Response

Dear Reviewer,

I would like to thank you first for providing such important suggestions and giving a recognizable value to this manuscript. Please kindly find the specific response (with line numbers) to your comments and also the revised manuscript based on your comments.

General comment from Reviewer-1:

Specific comments throughout the paper:

Affiliation

There is a single affiliation, with four email addresses. Please revise. 

Authors: This has been addressed. Please see lines 4-7 in the revised paper.

Abstract.

The abstract does not mention gold nanoparticles. There is a strike between the title and the abstract. Please revise the abstract.

Authors: This has been addressed. Please see lines 19-22 in the revised paper.

  1. Introduction

Line 28: POCT is defined in the abstract but not in the Introduction section text. Please fix.

Line 34: NP is not defined as an acronym. Please define it.

Lines 40-42: I suggest to number the list of limitations as i), ii) etc, or as a bulleted list. Indeed, there can be confusion, such as in the case of the "multiplexing capacity (<10), 2)". This is an editorial observation. 

Lines 43-47: Why listing explicitely the name of all these commercial tests? 

Lines 60-64: Missing references for these sentences. 

Lines 65-79: This part about COVID is appreciated and demonstrate that the manuscript is timely, while potentially being of interest to a large community of readers, such as that of Diagnostics. 

Authors: The above issues have been addressed. Please see lines 27,33, 39-42,_43-45, 79-82 in the revised paper.

Table 1 is really relevant for the readers' understanding. Thank you for providing it.

  1. Materials and Methods

The review methodology is explained. A point has to be clarified: in line 112, were the keywords for the search used in combination? This information should be reported. 

Authors: This has been addressed. Please see line 131 in the revised paper.

  1. Results 
    3.1 Lines 124-135: Missing reference(s). 

Authors: This has been addressed. Please see lines 144-162 in the revised paper.

Tables 2 - 5: I suggest to change the references from "(44)" to "[44]" in the first table column. There should be a coherent notation. 

Authors: This has been addressed. Please see tables 2-5 in the revised paper.

3.2 Lines 177-184: Missing reference for the MedMira product. Provide a website or a manual. 

Lines 185-188: Missing reference.

This section is strongly related to one company. The authors did not declare any conflict of interests, but the lack of comparison with other stakeholders and companies is not appropriate.

Authors: This has been addressed. Please see lines 198-204 in the revised paper.

3.3 Lines 218-219: Too many undefined acronyms. PLease fix. 

Authors: This has been addressed. Please see lines 238-249 in the revised paper.

3.5 The sub-subsections has to be better highlighted (title, font, etc.) for easing the readers' comprehension. 

Line 256: Extra "size". Please correct this typo. 

Lines 256-274: The text is not justified. Please revise the editing according to the template. 

Authors: This has been addressed. Please see lines 277-298 in the revised paper.

A note: Maybe some insight about the fundamentals of the physics and biochemistry of LFAs could be of help for the non technical readers who could read your review work and use it as start for their research (e.g., M.Sc. and PhD. students)

Authors: This has been addressed. Please see lines 46-64 in the revised paper.

Reviewer 2 Report

The systemic review manuscript is well organized with valuable information. The Diagnostic journal reader would get valuable information from this review. The current manuscripts need minor corrections before accept. The selective comment includes:

  1. The word's complete form needs to be written before writing a short form in every section—for example, POCT in the Introduction.
  2. Table 1, Table 2, and Table 3 need to add references as it is a review articles.
  3. Sections 3.1 to 3.5 belong to important information. It would be worth it and convenient if the author could add one schematic figure before the discussion chapter.

Author Response

Dear Reviewer:

Thank you for your kind comments. We have one question about #3 about adding one schematic figure before the discussion chapter. Could you please be more specific about what kind of schematic figure you deem convenient and worth for readers to better understand this review?

Sincerely appreciate your comments

Rongwei